# Protein control of photochemistry and transient intermediates in phytochromes

Giacomo Salvadori[1] ✉, Veronica Macaluso[1], Giulia Pellicci ⓘ [1], Lorenzo Cupellini ⓘ [1], Giovanni Granucci ⓘ [1] & Benedetta Mennucci ⓘ [1] ✉

Phytochromes are ubiquitous photoreceptors responsible for sensing light in plants, fungi and bacteria. Their photoactivation is initiated by the photo-isomerization of the embedded chromophore, triggering large conformational changes in the protein. Despite numerous experimental and computational studies, the role of chromophore-protein interactions in controlling the mechanism and timescale of the process remains elusive. Here, we combine nonadiabatic surface hopping trajectories and adiabatic molecular dynamics simulations to reveal the molecular details of such control for the *Deinococcus radiodurans* bacteriophytochrome. Our simulations reveal that chromophore photoisomerization proceeds through a hula-twist mechanism whose kinetics is mainly determined by the hydrogen bond of the chromophore with a close-by histidine. The resulting photoproduct relaxes to an early intermediate stabilized by a tyrosine, and finally evolves into a late intermediate, featuring a more disordered binding pocket and a weakening of the aspartate-to-arginine salt-bridge interaction, whose cleavage is essential to interconvert the phytochrome to the active state.

Phytochromes are light-sensing biological machines, ubiquitous in plants, fungi and bacteria[1–11]. They are red-absorbing homodimeric proteins, carrying a photoswitchable bilin as a chromophore. Most phytochromes exist in two distinct photoreversible forms: the red-light-absorbing form, also known as Pr state, and the far-red-light-absorbing form, the Pfr state[9,12–15], which differ by both chromophore stereochemistry and protein structure.

Phytochromes of different species share a generally conserved photosensory module (PSM, Fig. 1), composed by three domains PAS (Per/Arndt/Sim), GAF (cGMP phosphodiesterase/adenyl cyclase/FhlA) and PHY (Phytochrome specific). The PSM senses light and transfers the signal to an output module, which exerts a specific function depending on the species[16–18]. Owing to such a modular domain architecture, and because they absorb and fluoresce within the so-called "transparent window", phytochromes are exploited in emerging fields as bioimaging and optogenetics[10,19–25].

In bacterial phytochromes, such as the phytochrome from *Deinococcus radiodurans* (DrBph), a biliverdin IXα (BV) chromophore,

embedded in the GAF domain, is covalently bound to the protein through a cysteine residue (Cys24), belonging to the PAS domain (Fig. 1)[26–29]. In the resting Pr state, the "tongue" structural motif connects the PHY domain to the chromophore-binding pocket through a salt bridge involving a conserved aspartate, belonging to the GAF domain, and an arginine residue of the PHY domain (Fig. 1). In response to red light, the chromophore is electronically excited and photo-isomerizes at the $C_{15}=C_{16}$ double bond[30,31], initiating a cascade of structural changes, which propagate in the binding pocket and beyond. As a result of those changes, the tongue refolds from a $\beta$-sheet to a $\alpha$-helix, finally leading to the new, Pfr, conformational state of the phytochrome. Thanks to spectroscopic investigations, we know that the photocycle of bacteriophytochromes involves at least two intermediate states, namely Lumi-R and Meta-R[32–35]. The former arises directly from the initial photoisomerization; then, in tens of microsecond, thermal relaxation processes lead to the formation of the Meta-R intermediate. A further study[36] indicates that at least two Lumi-R intermediates are present in the DrBph photocycle: an *early* Lumi-R,

[1]Department of Chemistry and Industrial Chemistry, University of Pisa, Via G. Moruzzi 13, 56126 Pisa, Italy. ✉e-mail: giacomo.salvadori@phd.unipi.it; benedetta.mennucci@unipi.it

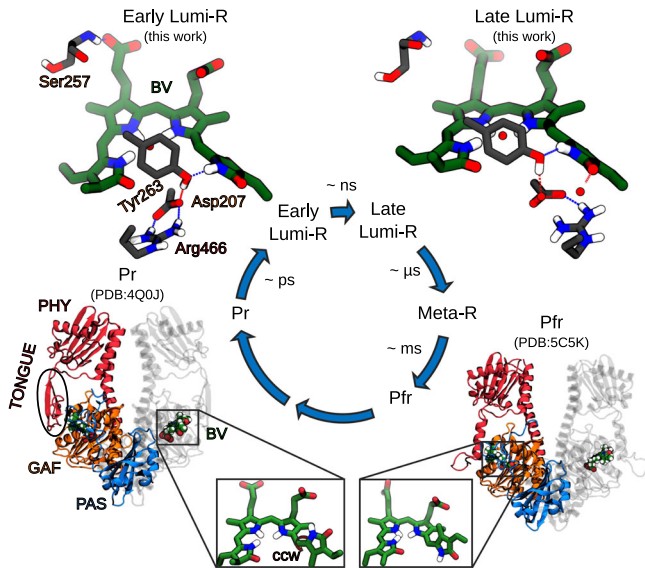

**Fig. 1 | The photocycle in *Deinococcus radiodurans* bacteriophytochrome.** Representation of the PSM of the two photoproducts: Pr (PDB ID:4Q0J https://www.rcsb.org/structure/4Q0J[27]) and Pfr (PDB ID:5C5K https://www.rcsb.org/structure/5C5K[28]), with a zoom on the bilin chromophore (green), highlighting the PAS (blue), GAF (orange) and PHY (red) domains. The D-ring counterclockwise rotation (ccw) of the chromophore is represented by the red arrow in the left inset[46]. Structures of the chromophore and the nearby residues obtained in this work for the early and late Lumi-R intermediates are reported in the two upper corners. The timescales for the different steps are the ones reported in the literature[36].

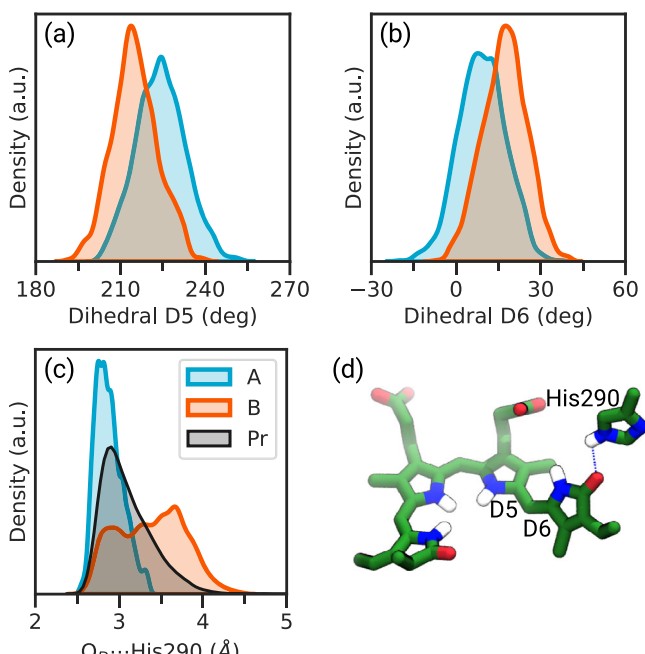

**Fig. 2 | Comparison of ground-state QM/MM trajectories of the Pr state.** **a**, **b** Distribution of the dihedrals D5 (**a**) and D6 (**b**) in the two sets of QM/MM-MD trajectories. **c** Distribution of the distance between D-ring carbonyl and His290 residue in the two sets of QM/MM-MD trajectories and in the resting Pr state. The Pr distribution has been generated from a 4 µs-long MM-MD simulation previously performed in our group[47]. **d** Representative structure of the BV in the Pr state with the His290 residue. Source data are available in the Zenodo repository[89].

which is formed on a ps-timescale after excitation, and a *late* Lumi-R, which is thermally reached from the former on a longer (ns) timescale (Fig. 1). Despite the numerous studies appeared so far[13,17,18,27,28,34,37–44], there is no consensus on the exact mechanism and the timing of the photoisomerization, neither we know the structures of the suggested intermediates and the molecular interactions that allow the propagation of the chromophore structural changes first to the binding pocket, and then to the entire protein.

Here, we use a multiscale computational approach to investigate the photoactivation of DrBph up to the microsecond timescale, revealing the molecular mechanism of the photoisomerization, individuating the intermolecular interactions, which lead to the Lumi-R intermediates and characterizing the corresponding structures. This investigation is made possible by the integration of non-adiabatic excited-state dynamics based on the mixed quantum-classical surface-hopping (SH) method[45] with adiabatic (ground-state) molecular dynamics simulations.

Our results show the presence of a conical intersection (CI) between the ground and the first singlet excited state of BV as a non-radiative decay channel. Such a CI is reached via a hula-twist mechanism, which involves a rotation of the D-ring in the counterclockwise direction according to the convention normally used in the literature[46] (see also the inset in Fig. 1). The resulting structure is here interpreted as the *early* Lumi-R intermediate. Notably, the mechanism of formation of this intermediate and the involved time are tightly connected to the type and strengths of interactions with close-by protein residues. In particular, we reveal the striking role played by the H-bond between the D-ring carbonyl oxygen and a conserved histidine, which affects the initial values of two BV dihedrals involved in the hula-twist mechanism and slows down the whole isomerization process. Moreover, after the CI has been reached and BV is back to the ground state, a new H-bond is formed between the D-ring and a tyrosine residue (Tyr263), in line with mutational studies[36,42]. The reliability of the resulting intermediate is validated by a direct comparison with the data from transient and step-scan IR spectroscopy[36,41]. Finally, we simulate the thermal relaxation into the *late* Lumi-R state, confirming that the latter is characterized by a more dynamic environment around the chromophore than the early intermediate. In particular, we observe a weakening of the GAF-PHY Asp207 ⋯ Arg466 salt-bridge interaction, whose cleavage is known to be essential to interconvert from Pr to Pfr.

## Results

The presentation of the results is divided in three parts. We first focus on the photoisomerization process. Then, we characterize the first intermediate reached from the photochemical process. Finally, we present the further evolution of this first (early) intermediate towards a second (late) intermediate.

### The photoisomerization

In a previous study, we investigated the conformational space of the resting (Pr) using a 4 µs-long MM-MD simulation[47]. From such a study, we confirmed a transient H-bond network within the binding pocket of the biliverdin chromophore as previously observed in other computational and experimental studies[34,48–54]. In particular, the hydrogen-bond network of the D-ring carbonyl group, which is expected to play a central role in the isomerization was found to mostly involve a neighboring water molecule, and partially, a histidine residue (His290). Before starting the surface-hopping (SH) simulation, we refined the description of the structure of the chromophore by running two ground-state QM/MM-MD trajectories (A and B in the following) performed as described in the Methods. The two trajectories showed a similar network of hydrogen-bond interactions, but in trajectory A we found a closer interaction between the D-ring carbonyl ($O_D$) and His290 (Fig. 2c). The stronger hydrogen bond with the His290 residue found in Trajectory A with respect to B is also reflected in

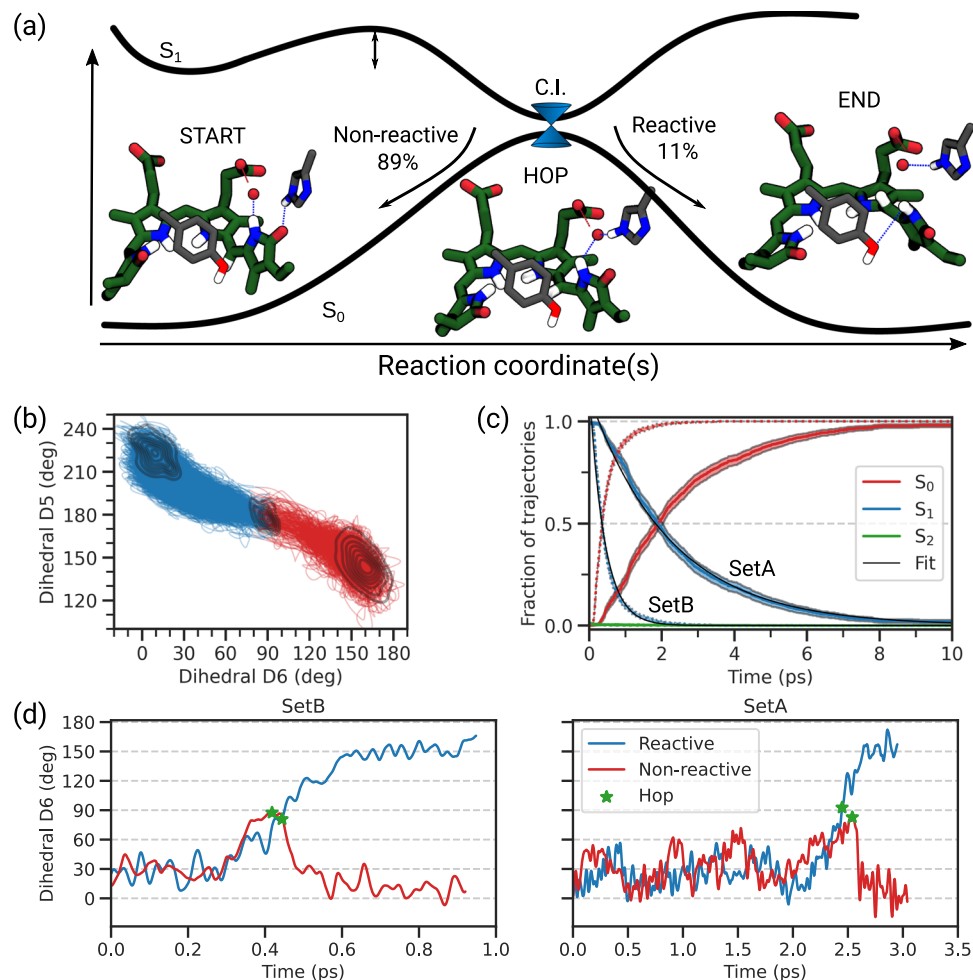

**Fig. 3 | The photochemical process. a** Sketch of the $S_1$ and $S_0$ PES along the reaction coordinate(s) with the quantum yield due to Set A. We represented the structures of the chromophore at the beginning of the simulation, at the time of the $S_1 \rightarrow S_0$ transition, and at the end of the simulation. The main hydrogen bonds involving the D-ring are highlighted. **b** Correlation between the dihedral angles D5 and D6. All reactive trajectories from Set A are shown using blue lines for trajectories running on $S_1$, and red lines for those on $S_0$. The density distribution is shown at the starting conditions, at the $S_1 \rightarrow S_0$ hop, and at the end of the simulation.

**c** Time evolution of the electronic state populations evaluated as the fraction of trajectories running on the given state at the given time for both Set A and Set B, which are the mean values, with their 95% confidence intervals. Black line: fit of the $S_1$ population according to an exponential function (Supplementary Methods 3). **d** Representative reactive and non-reactive SH trajectories from Set A and Set B. Green stars represent the $S_1$-to-$S_0$ transition. Source data are available in the Zenodo repository[89].

the distribution of the BV dihedral angles that are expected to be involved in the isomerization, namely D5 and D6 (Fig. 2a, b). A shift in the equilibrium values of D6 and D5 is observed in trajectory A with respect to B, which can be explained by saying that the stronger the hydrogen bond with His290, the more planar the D6 dihedral angle. A final comparison of the two QM/MM trajectories with the previous MM-MD simulation of the Pr state[47], shows that trajectory A describes the interaction between the carbonyl group and the nearby histidine more similarly to what found in the extended Pr sampling.

Configurations and momenta extracted from trajectories A and B were finally used as initial conditions for SH simulations, which were all initialized in the lowest $S_1$ excited state. For these simulations, we only considered the first three singlet states. In fact, it is known that in DrBph, triplet states are not involved in the photochemical process[36,41], contrary to what happens in the AnPixJ phytochrome, where triplet states are likely involved and the excited-state lifetime is in the nanoseconds timescale[55]. Overall, we analyzed 2442 SH trajectories, and we classified them as Set A and Set B based on the starting ground-state trajectories, respectively.

In both sets, the excited-state dynamics proceeds from the Franck-Condon region through a counterclockwise rotation around

the dihedral D6 towards a new conformation characterized by a D6 torsion of c.a. 90 degrees (Fig. 3b). The simultaneous clockwise rotation of D5 reveals a concerted isomerization around the double and single bonds (see supplementary Movie), also known as *hula-twist* mechanism, a space-saving isomerization inside a tight binding pocket[56,57]. Such a motion allows the chromophore to complete the isomerization around the double bond without a substantial rotation of the terminal moiety, which would be hampered by the steric constraints of the protein. This mechanism has been previously proposed for other conjugated systems, including photoactive protein chromophores[56,58].

The non-radiative decay from $S_1$ to $S_0$ proceeds through a conical intersection, reached when D6 is around 90 degrees. As shown in Fig. 3b the hopping geometries populate a small region in the dihedral space, where D6 is nearly 90° and D5 is almost planar (Supplementary Fig. 1). Representative structures of the Franck-Condon point and CI are compared in Fig. 3a.

After reaching the ground state through the CI, most trajectories return to the initial conformation, i.e., the Pr state (from here on, non-reactive trajectories), while a small population continues the rotation towards larger values of D6 (from here on, reactive trajectories). The

final geometry is represented in Fig. 3a. Reactive trajectories decay on average closer to 90° than the non-reactive counterparts (Supplementary Fig. 1).

As shown in the structures of Fig. 3a, the opposite motion of dihedrals D6 and D5 (Fig. 3b) causes the D-ring to rotate counterclockwise. Namely, the angle between the planes of rings C and D increases by c.a. 55 degrees on average. None of the trajectories (either from Set A or B) showed clockwise rotation as a possible reaction path. On the contrary, both reactive and non-reactive trajectories showed a similar initial evolution of dihedrals D5 and D6 (see Fig. 3b, d and Supplementary Movie). In many trajectories, the hydrogen bond between the D-ring carbonyl and His290 is lost upon reaching the CI region (Fig. 3a and Supplementary Fig. 2). In fact, the constraints imposed by the CI geometry, with the counterclockwise rotation of D-ring, do not allow an optimal interaction of the carbonyl with His290. In other words, His290 apparently stabilizes the Pr-like conformation of ring D over the CI geometry.

The time evolution of the excited-state populations is shown in Fig. 3c. The trajectories from Set A show an exponential decay with a time constant of 2.24 ps (Supplementary Table 1). Moreover, we could estimate a photoisomerization quantum yield of about 15% (Supplementary Table 1), which favorably compares to the experimental yield of formation for the Lumi-R state[48,54].

To assess the importance of the hydrogen bond with His290, we compared these results with the corresponding ones for the trajectories in Set B, which only marginally feature this interaction. Trajectories in Set B follow the same mechanism with counterclockwise rotation as Set A, but the excited-state lifetime is reduced to 0.48 ps (Fig. 3c and Supplementary Table 1). The reduced lifetime can be explained only by the fact that the CI geometry is reached earlier in Set B, after only a few oscillations around the initial value, while the trajectories in Set A spend more time around the Franck-Condon region (Fig. 3d). We deduce that the interaction between His290 and ring D creates an energetic barrier, which hinders the torsion around the double bond and slows down the attainment of the CI region.

In order to support this picture, we sampled additional ground-state QM/MM-MDs from different points of the Pr ensemble, and generated a total of 881 SH trajectories. These trajectories were pooled with sets A and B and categorized on the basis of the initial conditions (Supplementary Note 1). We find that the average excited-state lifetime is clearly correlated with the presence of a hydrogen bond to $O_D$ (Supplementary Fig. 3F). This hydrogen bond occurs either directly with His290 (e.g., what already observed on set A) or through a bridging water molecule.

To validate the findings here obtained about the role of chromophore-protein interactions in determining the mechanism and the timing of the isomerization, we report a comparison with the available literature.

The excited-state lifetimes are affected by the composition of the investigated phytochrome (i.e., chromophore-binding domain, photosensory module or full-length system)[50,59]. Moreover, a complex excited-state decay with multi-exponential kinetics has been observed in bacterial phytochromes[48,50,51,60,61]. Transient absorption measurements on the PSM of DrBph report an excited-state lifetime of about 170 ps[50], whereas transient IR experiments[62] have measured a 60 ps lifetime for the fastest decay component, and a recent femtosecond X-ray crystallography study[63] has detected a twist of the D-ring already after a delay time of 1 ps after photoexcitation. Our simulations, even for the slower Set A, show a deactivation process with a lifetime around 2 ps, significantly faster than transient absorption experiments. An even faster decay has been reported in a very recent computational study[64], using a different QM description and a different approach for simulating the excited-state dynamics of BV in DrBph: in that study 50 trajectories were generated of which 33 reached the conical intersection in less than a picosecond. Our SH trajectories show a

heterogeneous distribution of the energy gap between $S_0$ and $S_1$ states at the S1-to-S0 hop points. Therefore, to further assess the robustness of the calculated decay times we repeated set A and set B SH trajectories, forbidding hops when such a energy gap is larger than 0.5 eV. These simulations yield essentially the same results as the original dynamics, with slight differences in decay times and quantum yields (Supplementary Table 1), confirming that the excited-state decay passes through a CI region with small electronic energy gaps.

As we argued above, the achievement of the CI region is controlled by the energetic barrier on the torsional degrees of freedom. Therefore, a relatively small change in this torsional barrier can have substantial impact on the excited-state lifetime. To show how such energetic barrier to the torsion can affect the excited-state lifetime, we repeated the simulations on representative subsets of set A/B by artificially increasing the barrier around the dihedral D6, namely from the "real" one of 1.6 kcal mol$^{-1}$ to 5.2 kcal mol$^{-1}$ (Supplementary Fig. 4). In both sets, the reaction mechanism and the key interactions with the protein pocket remain unchanged. On the contrary, the lifetime increases exponentially with the height of the barrier (Supplementary Figs. 5 and 6), which is an indication of an activated process. We note that a recent study on cyanobacterial phytochrome (Cph1)[55] using an accurate QM method (CASPT2) has found a barrier of 6.2 kcal mol$^{-1}$ for the $S_1$ potential energy along the D6 rotation. If we assume that this estimate of the D6 barrier is valid also for BV in DrBph, an excited-state lifetime of the order of 100 ps is expected for Set A, a value in good agreement with experiments (Supplementary Fig. 6).

## From the photoproduct to the early Lumi-R intermediate

The latest conformations sampled by the SH reactive trajectories were used as starting point to characterize the time evolution of the photoproduct through ground-state QM/MM and full MM-MD simulations (see Methods).

These simulations indicate a stable structure of the chromophore, with dihedrals D5 and D6 remaining close to the values of the non-adiabatic simulations, whose distributions strongly deviate from the Pr state (Fig. 4a, b). On the contrary, all the other dihedrals remain nearly the same as in the Pr state. The only exception is represented by the

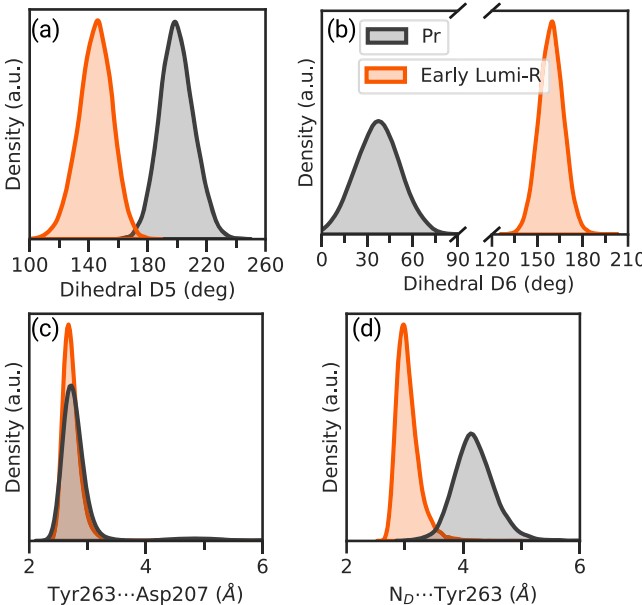

**Fig. 4 | Early Lumi-R intermediate. a, b** Distribution for the dihedrals D5 and D6, respectively; **c, d** Distribution for the Tyr263 ⋯ Asp207 and $N_D$ ⋯ Tyr263 distances, respectively. The distributions were made on the QM/MM (orange) and Pr (black) MDs. A representative structure of the early intermediate can be found in Fig. 1. Source data are available in the Zenodo repository[89].

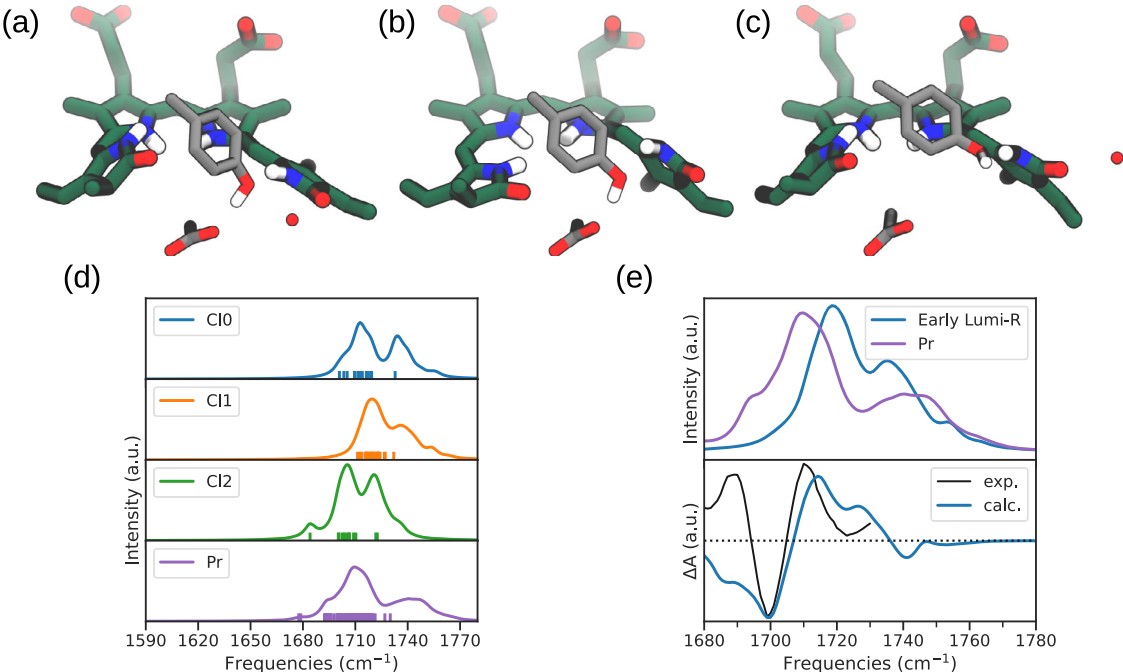

**Fig. 5 | IR spectroscopy characterization. a, b, c** Representative structures of cluster 0, 1, and 2, respectively. **d** IR spectra of the different clusters of Lumi-R and the Pr state. We have used 18, 26, and 13 configurations belonging to clusters 0, 1, and 2, respectively. The sticks represent the carbonyl $CO_D$ stretch. All spectra are normalized to the highest peak of Cl2. **e** In the upper panel, the averaged spectrum of the Lumi-R was represented together with that of the Pr state. In the lower panel, the theoretical (calc.) and experimental[36] (exp.) early Lumi-R − Pr difference spectra were represented (a shift of 8 cm⁻¹ has been applied to the calculated spectrum to properly compare with the experimental one measured in $D_2O$. Source data are available in the Zenodo repository[89].

first dihedral of the propionyl group attached to ring B (D1B, Supplementary Fig. 7), which is stabilized at Pfr-like values (Supplementary Fig. 8).

Extending the analysis to the binding pocket, we see that the hydrogen-bond network of the Pr state[65] is largely conserved with only one important exception. Our simulations suggest that Tyr263 stabilizes the Lumi-R state through a hydrogen bond with the D-ring nitrogen ($N_D$) in addition to the interaction with Asp207 already present in Pr (Fig. 4c, d). This finding is in agreement with mutagenesis studies[36,42], which have suggested that the Y263F mutation hinders the formation of Lumi-R, thereby reducing the photoconversion (Pr → Pfr) yield.

We conclude that the structural ensemble reached after photoisomerization is characterized by stable, but dynamic, hydrogen bonds of BV with the binding pocket. We assign this configuration to the *early* Lumi-R state. A representative structure is reported in Fig. 1.

To further validate such a structure, we have computed the IR spectrum and compared it with transient and step-scan IR spectroscopy experiments[36,41]. To select a set of representative configurations for the calculation of the IR spectrum, we have used a principal component analysis (PCA) based on intermolecular distances involving key residues in the chromophore-binding pocket and a hierarchical clustering algorithm. In this analysis we have combined the QM/MM trajectories and the first 10 ns of the MM-MD replicas. All the details on the clustering procedure and on the selection of the clusters are provided in Supplementary Note 2. Here, we note that the PCA analysis shows that the QM/MM-MDs explored a subset of the MM-MD configurations (see Supplementary Fig. 9). Three clusters, namely Cluster 0 (Cl0), Cluster 1 (Cl1), and Cluster 2 (Cl2), were identified (see Fig. 5a–c and Supplementary Fig. 9): Cl0 and Cl1 are very similar, but in the latter the hydrogen bonding between $CO_D$ and a water molecule is lost. Cl2 instead is characterized by a large distance between Tyr263 and Asp207 (Supplementary Fig. 10).

IR spectra have been calculated for each cluster and compared with the one calculated on the resting Pr ensemble (Fig. 5d). As the

experimental data indicate that the intermediate is characterized by a change in the frequency of the D-ring carbonyl with respect to the Pr state, owing to a different and dynamic local environment, here we focus only on the region of carbonyl stretches ($CO_D$ and $CO_A$ normal modes). For more details see the Methods section.

In Fig. 5d we can see how Cl0 and Cl1 give a rather similar representation, except for a shoulder in Cl0 at frequencies around 1700 cm⁻¹. Cl2, in which the Tyr263 ⋯ Asp207 hydrogen bond is absent, shows a major $CO_D$ peak at frequencies very close to the shoulder of Cl0. In fact, this signal is due to configurations in which a water molecule is hydrogen bonded to the carbonyl. On the contrary, configurations in which such interaction is absent show a peak at frequencies comparable to Cl1 (Supplementary Fig. 11).

An average spectrum was obtained through a weighted average of the three clusters (Fig. 5e) and compared with the one of the Pr state. In the average spectrum we have used the weights that reflect the populations of the three clusters in the QM/MM-MDs (0.18, 0.81 and 0.01 for Cl0, Cl1 and Cl2, respectively). The obtained difference spectrum (Lumi-R − Pr) has been finally compared with the experimental data (Fig. 5e, bottom panel). The calculated difference spectrum reproduces quite well both position and shape of the main positive and negative $CO_D$ signals. In particular, the correct reproduction of the frequency shift, experimentally observed when moving from Pr to the intermediate, further increases our confidence on the validity of the obtained structure. In fact, the difference spectrum is determined by a delicate interplay between the structural change in the chromophore and the modified chromophore-residues interactions. The good agreement with the experiment supports both structural details.

### From the early to the late Lumi-R intermediate

Having characterized the first Lumi-R intermediate, we now investigate the time evolution of the system in the μs timescale. Clearly, this is a time window that cannot be explored with QM/MM descriptions. For this reason, we have extended the 10 replicas used in the previous analysis to 1 μs each.

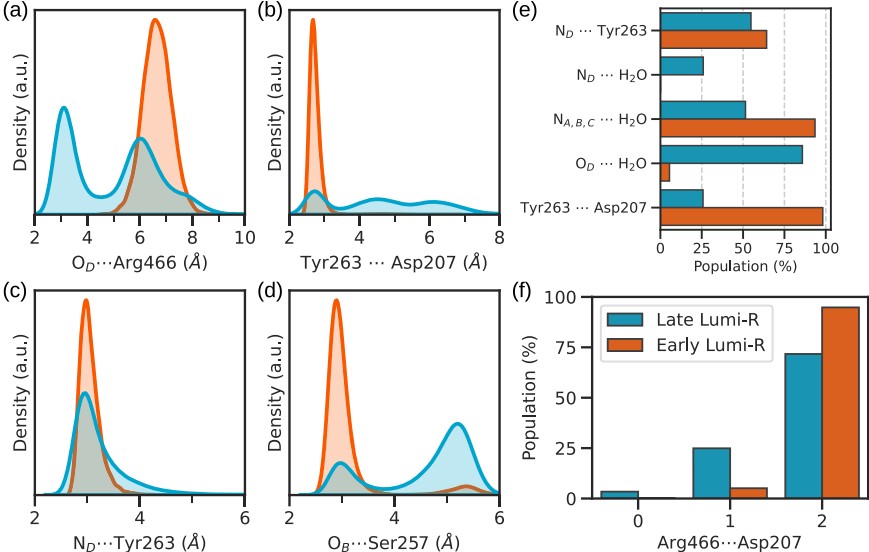

**Fig. 6 | Late Lumi-R intermediate. a–d** Distance distributions computed for the $O_D \cdots$ Arg466, Tyr263 $\cdots$ Asp207, $N_D \cdots$ Tyr263, and $O_B \cdots$ Ser257 interactions, respectively. **e** Probabilities for the main hydrogen-bonding interactions around the D-ring **f** Probabilities of having 0, 1, or 2 contacts between Asp207 and Arg466. We define a contact if the Asp207 oxygens, and Arg466 nitrogens are less than 3 Å apart. We compared the Early Lumi-R (in blue) against the Late Lumi-R (in orange). The Early Lumi-R analysis was performed on 20,000 frames, while the Late Lumi-R analysis was performed on 10,000 frames. For the Early Lumi-R, the error is less than $2 \cdot 10^{-4}$, while for Late Lumi-R it is less than $4 \cdot 10^{-3}$. A representative structure of the late intermediate can be found in Fig 1. Source data are available in the Zenodo repository[89].

Significant differences are found in the binding pocket, as shown by the larger RMSD fluctuations (Supplementary Fig. 12). In fact, multimodal distributions are observed for interaction between BV and close residues (Supplementary Fig. 13). Going into detail, some trajectories evolve towards a configuration in which Arg466 has an enhanced mobility and it can interact with the carbonyl D-ring (Fig. 6a). The distance between the D-ring carbonyl and Arg466 has a bimodal distribution, indicating a dynamic behavior of Arg466 (Fig. 6a), while in the *early* Lumi-R intermediate the narrow distribution indicates a rigid configuration. The dynamic behavior of this residue has been also confirmed by a recent study[41]. This observation is of considerable interest, as in the Pr state Asp207 forms a strong salt bridge with Arg466, whereas this contact is replaced by a hydrogen bond with Ser468 in Pfr.

The increased mobility of Arg466 has a knock-on effect on the residues around the D-ring: the salt-bridge interaction between Arg466 (belonging to the PHY domain) and Asp207 (belonging to the GAF domain), a hallmark feature of the Pr state, becomes weaker (Fig. 6f), and the interaction between Asp207 and Tyr263 becomes less frequent (Fig. 6b). The former represents a crucial element in phytochromes photoactivation, since it is needed to keep the tongue in its Pr-like form, and has to be released to reach the final Pfr state. Furthermore, a water molecule shields the interaction between Asp207 and Tyr263, causing its cleavage and the opening of the "cage", formed by a hydrogen-bonding network among Arg466, Asp207 and Tyr263 (Fig. 1), with the final exit of the pyrrole water (Fig. 6e). As the Asp207–Tyr263 bridge weakens, the hydrogen-bonding interaction between the D-ring and Tyr263 becomes weaker as well (Fig. 6c), and the pyrrole water takes the place of Tyr263 in the interaction with the D-ring (Fig. 6e). Therefore, according to our simulations, the pyrrole water is not ejected during the photochemical event, but when the Asp207 $\cdots$ Tyr263 interaction is broken. Moreover, the hydrogen bond between B-ring propionate and Ser257 becomes weaker (Fig. 6d), while the one with Tyr216 is strengthened (Supplementary Fig. 13).

All these changes in the intermolecular interactions and the resulting structural modifications with respect to the previous intermediate indicate that a new stable structure has been reached. This structure remains stable in all the μs-long trajectories. On the basis of these characteristics and the involved timescale of formation (from 10 ns to 1 μs), we assign this new structure to the *late* Lumi-R intermediate[36]. A representative structure is reported in Fig. 1.

## Discussion

DrBph is a prototypical bacterial phytochrome in which the thermodynamically most stable (resting) state is the Pr form, and red-light absorption is needed to trigger the conformational switch towards the Pfr active state. In other phytochromes, known as bathy, the stability is reversed, with Pfr being the thermally stable dark-adapted state. However, these different behaviors are not due to the chromophore but instead to the specific composition and shape of the binding pocket[66–68].

A comprehensive understanding of the photoactivation mechanism of phytochromes is still far from being achieved. In particular, an a priori knowledge of how the environment affects the entire activation process is still an open question. Here, by combining excited-state surface-hopping simulations and ground-state MD simulations, we have revealed the mechanism through which the binding pocket controls both the reaction mechanism and kinetics of the photoactivation of the Pr state in DrBph[69]. In particular, the compact structure of the protein pocket forces the chromophore to proceed through a concerted rotation around two adjacent dihedrals, a mechanism known as hula-twist motion, leading to a counterclockwise rotation of the D-ring. This finding seems to disagree with a previous study based on CD experiments[46], which proposed clockwise rotation as the most probable mechanism. However, the clockwise mechanism was suggested on the comparison of the initial and final states (Pr and Pfr) and on steric and chromophore-confinement arguments, without taking into account the conformational dynamics of the system or the role of the protein environment. Conversely, a counterclockwise rotation was directly observed in time-resolved X-ray experiments[63,70], which are consistent with the timescales simulated here. Moreover, also the previously quoted computational study reports a very similar mechanism to the one found here confirming both the hula-twist and the counterclockwise results[64].

According to our results, the direct (through hydrogen bond) or indirect (through a water molecule) interaction with a widely

conserved histidine (at position 290 in DrBph) controls the photoisomerization by slowing down the photochemical process. The importance of this residue has been also seen through mutational analysis of DrBph: its replacement with Gln blocks the Pr-to-Pfr photoconversion to the Meta-R intermediate[34] and with Thr reduces the extent of Pfr formation[62]. The importance of such residue is spread out to the entire phytochrome superfamily: when the conserved histidine is mutated to alanine in the bathy phytochrome PaBphP, the mutant adopts the Pr dark state and photoconverts to the Pfr state, i.e., it becomes a canonical bacteriophytochrome[71]. Moreover, in another prototypical phytochrome from the myxobacterium *Stigmatella aurantica* (SaBphP1), where His is naturally substituted by a Thr residue[72], a much faster excited-state decay is found together with a significantly higher Lumi-R quantum yield compared to DrBph[62].

One may wonder what are the design principles underlying slow photoisomerization with low quantum yields. We first note that low photoactivation yields are not uncommon in photoactive proteins, such as the orange carotenoid protein[73]. In addition, the protein needs to energetically stabilize the inactive state, in this case the Pr form, in order to ensure a complete thermal reversion in the dark. Finally, the residues in the binding pocket contribute to steer the entire photocycle towards the active state. Therefore, the identity of the residues composing the binding pocket is the result of a balance between photoisomerization efficiency and specificity of the entire photocycle.

As a matter of fact, our simulations confirm that the binding pocket has a fundamental role on the chromophore dynamics even beyond the initial photoisomerization process. In particular, we find a strong indication that the protein can guide the further evolution of the system towards the correct final state. Both Lumi-R intermediates, *early* and *late*, are stabilized through an interaction with Tyr263. Such a residue is essential for proper Pr-to-Pfr photoconversion. Indeed, when it is replaced with a histidine residue, the mutant fails to properly photoconvert to Pfr[34,36].

The evolution of the early intermediate has been followed via µs-long trajectories and shown to reach the *late* intermediate, which is characterized by a much more disordered binding pocket where the Arg466 ⋯ Asp207 salt bridge, a hallmark feature of the initial Pr state and still strong in the *early* Lumi-R, is significantly weakened. Such salt bridge has a central role in unleashing the activation pathway of the phytochrome, since it is needed to keep the tongue in its Pr-like form, and has to be released in order to biologically activate the protein. Thus, at a longer timescale, we can expect the detachment of Arg466, allowing the tongue refolding and the formation of the Pfr state.

According to our study, biliverdin in the late Lumi-R intermediate presents a structure that is in between those found in Pr and Pfr states when we consider the D6 and D5 dihedral angles. This finding suggests that the final (Pfr-like) structure of BV will be reached together with the other structural changes of the protein on a longer timescale. A recent computational study[64] instead proposed a late Lumi-R structure presenting the chromophore with the configuration of the Pfr state, while the protein is still in the Pr conformation. We note however that such a structure was obtained through a biased MD simulation using as the driving collective variable the BV dihedral angles. In our simulations instead, no biases have been used, but the system is allowed to follow its "natural" dynamical evolution in the µs time window. This should represent a more realistic modeling of the early-to-late transformation path. Moreover, the spectral differences experimentally observed between early and late Lumi-R states are rather small and mostly involve the nearest residues/water molecules surrounding the carbonyl group of the D-ring[36] as we find in our simulations. Finally, the early Lumi-R state has been shown to thermally relax to the late Lumi-R state with a lifetime of about 0.4 µs[36,41], which is perfectly in line with our predicted timescale.

The results here obtained represent a detailed atomistic characterization of the ps-to-µs steps of the complex transformation leading from the Pr to the Pfr state of the *Deinococcus radiodurans* bacteriophytochrome. It is important to emphasize that the methodology here used has allowed to connect the very different timescales of the process, from the ultrafast one involved in the photoisomerization to the one characterizing the rearrangement of the binding pocket, in a consistent way, without introducing any a priori assumption on reaction mechanisms or artificial driving forces. Because of these characteristics, we expect that the same strategy will allow investigating the photoactivation of other photoreceptors.

## Methods

### Sampling of the Pr state
In a previous study, we investigated the conformational space of the resting (Pr) using a 4 µs-long MM-MD simulation[47]. Five different configurations were extracted from such dynamics and used as starting points for ground-state QM/MM-MD trajectories. The latter were run for 20 ps starting using a time step $\Delta t = 0.1$ fs and the Bussi-Parrinello stochastic thermostat at 300 K. To keep the computational cost feasible, QM/MM trajectories were run on a reduced system (Supplementary Fig. 14) and to avoid any evaporation of water molecules, we added a constraining potential. The BV chromophore was treated with the multi-reference floating occupation molecular orbital-complete active space configuration interaction (FOMO-CASCI) method[74,75] in combination with the AM1 Hamiltonian. An active space of (6,6) was used in combination with Gaussian width of 0.05 a.u. for FOMOs (details on the selection of the QM method are provided in the Supplementary Methods 1). Conversely, the protein was described with the AMBER ff14SB force field[76] and water molecules with TIP3P. The boundary between the covalently linked QM and MM parts was treated using a connection atom scheme previously used for semiempirical QM/MM approaches[77]. According to this scheme, the $C_{3\beta}$ carbon atom of the chromophore is the connection atom, which behaves as a hydrogen atom in the QM part and as a normal carbon atom in the MM part (Supplementary Figs. 7 and 14). The QM/MM trajectories were performed with a modified version of the semiempirical program MOPAC2002[78], interfaced with the TINKER software package[79].

### Surface-hopping simulations
The last 10 ps of the five ground-state QM/MM trajectories were used to extract the initial conditions (nuclear coordinates and momenta) for generating the swarm of non-adiabatic surface-hopping (SH) trajectories. In particular, two trajectories, A and B, were used to generate a total of 2442 SH trajectories classified as set A and set B, respectively. The other ground-state trajectories were used to generate 881 additional SH trajectories for an analysis of the H-bonding effects (Supplementary Note 1). In all SH trajectories, the QM and the MM subsystems, as well as their respective levels of theory, were the same used in the QM/MM ground-state simulations. The local diabatization algorithm[74,75] was used for the integration of the time-dependent Schrödinger equation for the electrons, and quantum decoherence was taken into account by applying an energy-based correction[80]. More details are provided in the Supplementary Methods 2. SH simulations were performed with a modified version of the semiempirical program MOPAC2002[78], interfaced with the TINKER software package[79]. All the figures, the analysis of the trajectories, and the fitting were performed with matplotlib[81], in-house FORTRAN 90 scripts, SciPy[82], and in-house Python scripts.

### MD simulations of the intermediates
Configurations of the photoproduct obtained from the last frame of the reactive SH trajectories were used to propagate adiabatic MD simulations of the system in the ground state in search of the Lumi-R intermediates. In these simulations, we reintroduced a full solvation of the protein using a truncated octahedron water box. Both QM/MM and MM-MD simulations were performed. In the former case, for the chromophore we used the same semiempirical method (AM1) of the previous

non-adiabatic SH simulations. Four QM/MM trajectories were propagated for 5 ns each and average-quantity analysis were made on 5000 frames per trajectory. The full MM-MD simulations were instead used to extend the investigation of the intermediate evolution in the μs-timescale. For the chromophore and the covalently bonded cysteine residue Cys24 we used the same GAFF parameterization successfully used in a previous work on both the (resting) Pr and (active) Pfr states of the phytochrome[47,65]. Ten independent replicas were run for 1 μs each and average-quantity analysis were made on 1000 frames per trajectory. For both QM/MM and MM-MDs, the protein and water were described with the same force field used in the excited-state simulations. All MD simulations were run with AMBER18[83,84]. More details are provided in the Supplementary Methods 4. All the figures, the analysis of the trajectories, the featurization, dimensionality reduction and clustering were performed with CPPTRAJ[85], scikit-Learn[86], matplotlib[81], seaborn, and in-house Python scripts.

## Calculation of the IR spectra of the early intermediate

To simulate IR spectra of the early Lumi-R intermediate, we extracted 57 frames from the adiabatic QM/MM-MDs. For each frame, we considered the protein and a shell of 390 water molecules centered on the chromophore and we performed a geometry optimization of the QM subsystem in a frozen environment represented as an electrostatic MM embedding. The QM part included the BV chromophore, the side chains of Cys24, His290, and Tyr263, and the closest water molecules within 3.2 Å of ring D carbonyl and amidic nitrogen. The Cys24 residue was cut at the $C_A$-$C_B$ covalent bond (Supplementary Fig. 7). As QM level we used B3LYP+D3 in combination with 6-311G(d,p) basis set. For these calculations, we used the ONIOM(QM:MM) scheme[87], implemented in the Gaussian 16 suite of programs[88]. More details about the calculation of the IR spectra are reported in the Supplementary Methods 5.

## Reporting summary

Further information on research design is available in the Nature Portfolio Reporting Summary linked to this article.

## Data availability

The data supporting all the findings of this study are available within the article and its supplementary files. The source data generated in this study have been deposited in the Zenodo repository (https://zenodo.org/record/7254721). The PDB structures analyzed in this study can be accessed at https://doi.org/10.2210/pdb4Q0J/pdb(Pr) and https://doi.org/10.2210/pdb5C5K/pdb(Pfr). The source data that support the findings of this paper are provided with this paper. Source data referring to all the graphs reported in Figs. 2–6 are provided with this paper. Additional data, including those underlying all the Supplementary Figures, and Supplementary Tables, are available in the Zenodo repository (https://zenodo.org/record/7123603).

## Code availability

The custom code used for this study is available from the corresponding authors upon request.

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

## Acknowledgements
V.M., L.C., B.M. acknowledge funding by the European Research Council, under the grant ERC-AdG-786714 (LIFETimeS).

## Author contributions
B.M. acquired funding; G.S. performed the SH simulations; G.S., V.M., and L.C. performed the MD simulations and data analysis; G.S. and G.P. performed the QM/MM IR calculations; G.S., L.C., G.G., and B.M designed the research; All authors wrote and edited the paper. All authors approved the final version of the paper.

## Competing interests
The authors declare no competing interests.
