## [Peer Review File · Nature Communications]

REVIEWER COMMENTS

Reviewer #1 (Remarks to the Author):

The authors present a valuable piece of work that I recommend for publication in Nature Communications after revision. More specifically, the work reports on the investigation of the key mechanistic elements driving the relaxation of the chromophore-protein region on a microsecond time scale. This is, in my view, original, important and properly done. I particularly liked the description of the role played by the Tyr263 in, possibly, weakening the pre-existing salt bridge between Arg466 ... Asp207. Such a mechanism is of general interest in biophysics.

Specific comments:

Abstract

Given the level of Nat. Comm. the authors are invited to stress what the novelty is in their research that has never been previously addressed computationally.

Main text, page 3.

1. It seems that there is a strong dependence of the trajectories A and B on the starting configuration (or initial conditions). See also the dramatic difference in Fig. 3c. It is therefore rather surprising that only two trajectories have been computed. The authors should explain carefully the reason for not performing a more exhaustive characterization/sampling of the Pr state. There is presently no explanation.

2. This is a minor note but it may be relevant. When following the evolution of the dihedral angles, It would be important to also look at the pyramidalization of the carbon centers (especially the carbon shared by the two twisting bonds). In fact, while the dihedral angles appear a natural choice their value is strongly affected by the pyramidalization of the three centers forming the reactive framework. In other words, one can get large dihedral angle values not because of the twisting but because of the pyramidalization.

3. Fig. 2, legend. It is not clear how the the Pr distribution has been generated in panel c.

4. Has the CI been optimized and characterized for some configurations and, thus, studied more carefully (e.g. is that a peaked or sloped CI)?

Main text, page 4.

5. The unidirectional isomerization motion is caused by the chirality of the protein environment and it is reminiscent of the well know purely unidirectional motion documented in certain rhodopsin proteins (especially bovine rhodopsin). However, I ask the authors to be careful and give their definition of counterclockwise and clockwise. In fact, such definition depends from the reference or convention adopted. As a matter of fact when adopting the convention adopted for bovine rhodopsin the present motion would be clockwise. This may confusing a reader not familiar with the specific convention used for bilin.

6. "...the type of construct..." -> "...type of computational model...."

7. "...To assess the robustness of the obtained decay times, we firstly repeated set A and set B SH trajectories, forbidding hops when the energy gap between electronic states S0 and S1 is larger than 0.5 eV....". It is not easy to understand the significance of this. Wouldn't be enough to plot in an histogram the distribution of the energy gaps at the hop points?

8. "...of set A/B by artificially increasing the barrier around the dihe- dral D6, namely from the "real" one of 1.6 kcal mol⁻¹ to 5.2 kcal mol⁻¹ (Fig. S10)....". How is this barrier computed? Did the authors perform TS search or just a relaxed scan?

9. Fig. S12. The authors have to state which kinetic theory has been used to compute the mentioned lifetimes: TST? That would require to assume the thermal equilibration of a planar excited state.

10. "...the final ground state configurations of the SH reactive trajectories...". How is this configuration exactly defined?

Main text pag. 5.

11. Fig. 3a. This is a minor point. It is not clear why the authors depict the conical intersection reagon with an avoided crossing (this is confusing). That would be OK if a real minimum would exist in the vicinity of a conical intersection (i.e. if the conical intersection would be sloped rather than peaked).

12. "... the propionyl group (D1B)h...". This has to be indicated in some of the chromophore schemes above.

Main text pag. 8

13. "Discussion" -> "Discussion and Conclusions"

14. "... The different behavior is not determined by the type of chromophore but instead by the specific composition and shape...". This conclusion appears to be too strong. Is there a solid evidence for the irrelevance of the chromophore type and structure? When dealing with biological photoreceptors this type of statements are generally too strong to be true. Biological evolution has usually evolved both the chromophore and the cavity composition to optimize a certain function. These "parts" are mechanisms of a single machinery. I would modify these strong statements to remain on the safe side.

Main text pag. 9

15. "... As a matter of fact, our simulations confirm that the binding pocket has a fundamental role even beyond the initial photoisomerization process...". Again, the authors should not stress this factor which is a trivial one (just think to the color tuning in different photoreceptors, as well as the differences in behavior with respect to the solvent environment). Of course, what is really important here is the description of the actual molecular level interactions bringing about a certain behavior especially after the isomerization. This is, in a sense, a novelty in the field as it is not often explored especially on the microsecond timescale.

16. "... see Fig. S12...". There is a mismatch between the figure number and the actual reported data. I did not find such data in the Supporting Information.

17. "...According to this scheme, the C3b carbon atom of the chromophore is the connection atom which behaves as a hydrogen atom in the QM part and as a normal carbon atom in the MM part (Fig. S7)....". Is this just the standard H-link atom method?

Main text, page 10.

18. "... were initially run for 20 ps using a time step $\Delta t = 0.1$ fs and the Bussi-Parrinello stochastic thermostat at 300K. The last 10 ps were used for sampling the initial conditions (nuclear coordinates and momenta) ...". 20+10 ps is really far too short for a complete equilibration run. I guess that this

is due to the fact that the authors use a QM/MM method that is probably (even if a semiempirical QM method is used) much more expensive than the MM method. A different strategy adopted in the literature would have been to equilibrate the system with MM for several hundred ns using a parametrized chromophore and then apply the QM/MM equilibration starting from the MM ns-equilibrated system. I wonder why this strategy has not been adopted.

19. The set up of the the A and B runs of Pr needs to be described in more details.

20. "...was used for the integration of the time-dependent Schrödinger equation for the electrons, and quantum decoherence was taken into account applying...". One sentence defining the SH method used needs to be inserted here.

21. "...As now the only investigated electrostatic state is the ground state, an HF/AM1 approach has been used to describe the chromophore in the QM/MM MD simulations....". The authors choose to use a semiempirical QM method rather than parametrized the chromophore. A justification/motivation for this shall be introduced here.

22. "...There, the GAFF force-field was used for BV and for the bonding interactions between BV and the covalently bonded cysteine residue Cys24...". Is the GAFF force-field good enough? The chromophore could, in principle, be specifically parametrized. I wonder if this has been done by other groups working on phytochrome chromophores.

23. "...On such subsystem, we performed a ONIOM(QM:MM)82 geometry optimization, in which the QM part was treated at B3LYP+D3/6-311G(d,p) level of theory...". This is a kind of awkward procedure. Initially the QM/MM model used by the author was based on the link-atom scheme and therefore an additive scheme. Now, suddenly, the author switch to a subtractive scheme. This is not justified. There is I think not good reason for this switching of "technology". So, an explanation has to be provided. I mean I understand that for computing IR spectrum one has to improve the QM level of theory. However this can be done also within the consistent link-H atom additive scheme.

Reviewer #2 (Remarks to the Author):

The paper "Protein control of photochemistry and transient intermediates in phytochromes" by Salvadori et al reports QM and MM simulations on a canonical bacteriophytpchrome. The work covers the photoisomerization process and the first ground state intermediate (lumi-R). The authors report evidence for how the chromophore isomerizes, how the protein environment controls this isomerization, and how the protein and chromophore develop the signal in the lumi-R state. The report underpins a recent result (by IR spectroscopy) that there are two lumi-R states present. I find these results important, because they connect atomic structure, photochemistry and signaling in phytochromes, which are a widely debated topic and because it advances our knowledge in the area.

Although some of the ideas had already been mentioned as ideas and models by others, this is the first rigid theoretical investigation that sensibly explains the formation of the lumi-R state. The paper gives a very plausible explanation at the atomic level on how bacteriophytochromes get photoactivated. It has the potential to become a paradigm model for phytochromes for many years to come and is therefore of highest interest for the phytochrome community.

The work advances the understanding of how proteins control (photo)chemistry. This is relevant in many more fields: biochemistry, photosynthesis, and related fields.

Although I have some minor comments below, I am pleased with the current state of the paper. I

believe that the simulations support the conclusions and claims. I am not an expert in QM/MM, and cannot judge the technical validity of the simulations, but judged against basic thermodynamic and photochemical knowledge, the conclusions are well supported by the data. The conclusions also align well with previous literature. The methodology and choice of simulations is sensible and appropriate. The work should be reproducible from the details provided. I recommend publication with minor revisions.

Comments:

Fig 1: I find it hard to understand which parts of the figure come from the paper and are new and which not. Could you clarify/revise the figure? It would be good to indicate what is "this work" and what is "from literature".

Time scales: Fig 3 is used to claim time scales of the transition. Could you discuss the validity and certainty of these time scales? Surely this must depend on the choice of force field/basis set/resting state structure? Spectroscopy indicates a long excited state lifetime of up to nanoseconds for the DrBpHP. This is now observed in the simulations. Could you discuss this?

Related: How accurate is the determination of the time scale of transition from early to late Lumi-R? It appears to agree with the experimental value, but is the accuracy of the time constant not quite inaccurate from the MD simulations?

Fig 4 and 6 - please indicate structure of the late and early intermediates. This would help to understand the figures.

The absence of any protonation/deprotonation reactions in the model deserve discussion. Heyne has recently demonstrated fast deprotonation in the Pfr excited state; Hildebrand, Mrogrinski escobar; Ihalainen; and others have a number of recent papers indicating the importance of a proton equilibria in the binding pocket, so protonation is seems to be important. That it is not observed in the simulations of the authors could be a limitation of the model. Could you discuss this issue?

Could you add some discussions regarding the (potential) limitations of the force field with respect to the lumi-R structures? To me it seems that especially the torsion angles of the BV are strongly dependent on the choice of dihedral constraints. Groenhofs JPCL 2022 recent paper does not arrive at the same structure for Lumi-R and it is said that the present simulations are unbiased, whereas Groenhofs are not - but is this really true give that there always is a force-field involved in MD simulations? I think it is important to discuss the potential limitations for the method.

Reviewer #3 (Remarks to the Author):

Phytochrome are a class of photoreceptor in plants, bacteria and fungi used to detect light. The initial photoisomerization process will result in the structural changes from the chromophore to the whole protein. Giacomo et al. tried to uncover the molecular mechanism for how the interaction between the chromophore and the protein residue controls the sequence structural changes. They use an integrated multiscale molecular dynamic simulations method and found that the photoisomerization of the chromophore proceeds through a hula twist mechanism. The results in this paper may give rise to the interest of related areas, and my detailed comments are as follows:

1. In Fig. 2, they ran two sets of ground-state QM/MM MD trajectories denoted as A and B, respectively. Then, they just believe that the trajectory A better captures the distribution of the dihedrals of D5 and D6. However, this inclusion is insufficient to acquire without further simulation sets. They should run at least two or more sets to double-check their results.

2. The detailed simulation system is ambiguous. Especially, whether the system contains water molecules around the chromophore. When water molecules are taken into account, the transition energy $S_0 \rightarrow S_1$ and the PES of the excited state S_1 may be altered. In addition, the photoisomerization starts configuration with a hydroxyl group which does not create the hydrogen bond. Is this hydroxyl group hydrogen-bonded to the water molecule?

3. The excited-state lifetime was just estimated by the time evolution of the electronic state population. This method may be random in statistics, which strongly rely on the initial setting. Moreover, quantitatively measuring this value usually requires averaging over many sets of electronic excited state simulation. Can they thetically get a lifetime through the stimulated emission equation?

4. The whole calculation processes are rather complicated. For example, they do ground-state QM/MM with two sets first. Then, they extracted selected trajectories from these trajectories to use as the beginning setup for the exciting computation. In addition, specific trajectories must be chosen for the IR computations. Can the authors include a schematic diagram to clarify? The authors may also show that all the vibrational frequencies are positive for the IR simulations in the SI.

5. They calculated three single states but did not display the corresponding absorption spectrum. I am unable to assess the oscillation strength of $S_0 \rightarrow S_1$. As a result, they must evaluate a significant number of single states, such as twenty. This will provide additional information to determine how the photo is relaxed to the S_1 . Furthermore, the authors need to perform another excited state dynamics simulation, in which the initial absorption state is from S_0 to S_2 .

6. In the surface hopping trajectories, they use the semiempirical method (AM1). Also, they have performed a benchmark against the method of TDCAM-B3LYP. Even so, the red line (TDDFT) in Figure S3 is incomplete, in contrast to the blue line (AM1). They should run a test of surface hopping trajectories in the framework of TDDFT, in order to confirm the feasibility of AM1 in this system.

7. In the abstract, the mapping method description into method content reflect that the using method is too abstract to understand. They should descript the method with a more explicit statement, such as surface hopping dynamics simulation of AM1.

Reviewer #1 (Remarks to the Author):

The authors present a valuable piece of work that I recommend for publication in Nature Communications after revision. More specifically, the work reports on the investigation of the key mechanistic elements driving the relaxation of the chromophore-protein region on a microsecond time scale. This is, in my view, original, important and properly done. I particularly liked the description of the role played by the Tyr263 in, possibly, weakening the pre-existing salt bridge between Arg466 ... Asp207. Such a mechanism is of general interest in biophysics.

Authors' Reply: We sincerely thank the reviewer for the positive comments but even more for the critics and suggestions which have allowed us to significantly enhance the paper's impact and clarity.

Specific comments:

Abstract

Given the level of Nat. Comm. the authors are invited to stress what the novelty is in their research that has never been previously addressed computationally.

Authors' Reply: As suggested we have revised the abstract to clearly report the main novelties.

Main text, page 3.

1. It seems that there is a strong dependence of the trajectories A and B on the starting configuration (or initial conditions). See also the dramatic difference in Fig. 3c. It is therefore rather surprising that only two trajectories have been computed. The authors should explain carefully the reason for not performing a more exhaustive characterization/sampling of the Pr state. There is presently no explanation.

Authors' Reply: We thank the reviewer for the question. Indeed, we fully agree that this point (also touched by Reviewer #3) is an important one and, in fact, to further support our conclusions we have extended the analysis as detailed here below. However, before describing the new results we would like to clarify the computational protocol used in this work that, we admit, was not very clearly described in the original version of the manuscript.

In a previous study (ref. Phys. Chem. Chem. Phys., 2020, 22, 8585-8594), we investigated the conformational space of the resting (Pr) through a 4 μ s-long MM MD. Before starting the surface hopping simulations, we refined the description of the chromophore's geometry by running two 20 ps-long ground-state QM/MM MD trajectories (A and B in the manuscript) starting from two different configurations extracted from the μ s-long MM MD. The last 10 ps of the QM/MM trajectories A and B were then used for sampling the initial conditions (nuclear coordinates and momenta) for the swarm of SH trajectories. A total of 2442 SH trajectories were generated and classified into two sets, set A and set B.

Following the Reviewer's suggestion, we have now performed 3 additional 20ps-long ground-state QM/MM MD simulations starting from different configurations of the μ s-long MM MD. Then, we generated 881 additional SH trajectories using the last 10 ps of the three new QM/MM MD simulations. We then combined all the trajectories (set A, set B, and the new 881 trajectories) and classified

them based on the interactions between the D-ring and the closest residues at the starting geometry. By using a hierarchical agglomerative algorithm, we obtained 4 clusters. The most populated cluster 0 is characterized by a hydrogen bonding distance between D-ring carbonyl (O_D) and His290 and shows long lifetimes (this cluster contains the original set A). Instead, Clusters 1 and 2, where such interaction is weak or absent, have much shorter lifetimes (like Set B). Cluster 3, even though His290 is not at hydrogen bonding distance to O_D , shows excited state lifetimes of the order of Cluster 0. This behavior can be explained by noting that in this cluster, O_D is H-bonded to a water molecule, which also interacts with His290. We think that these further simulations highlight even more strongly the role of hydrogen-bonding interactions at O_D in determining the lifetime. These results are now reported in the Supplementary Material.

In the main text we have also revised the “Methods” section (see below) and the beginning of the subsection “Photoisomerization” in the Results section, and added the following comments (page 4):

“... In order to support this picture, we sampled additional ground-state QM/MM MDs from different points of the Pr ensemble, and generated a total of 881 SH trajectories. These trajectories were pooled with sets A and B and categorized on the basis of the initial conditions (Supplementary Note 1). We find that the average excited-state lifetimes clearly correlated with the presence of a hydrogen bond to O_D (Supplementary Fig. 4F). This hydrogen bond occurs either directly with His290 (e.g., what already observed onset A) or through a bridging water molecule.”

2. This is a minor note, but it may be relevant. When following the evolution of the dihedral angles, it would be important to also look at the pyramidalization of the carbon centers (especially the carbon shared by the two twisting bonds). In fact, while the dihedral angles appear a natural choice their value is strongly affected by the pyramidalization of the three centers forming the reactive framework. In other words, one can get large dihedral angle values not because of the twisting but because of the pyramidalization.

Authors’ Reply: As suggested by the reviewer, we looked at the pyramidalization of the carbon shared by the D5 and D6 dihedrals. We have defined the pyramidalization as the angle between the sum vector ($v_1 + v_2$) and the vector v_H (see below)

As it can be seen from the figure below, At the beginning of the nonadiabatic trajectories, the pyramidalization is essentially null.

At the time of the S1-to-S0 hop (vertical dotted line), there is a decrease (by a few degrees) but after that, the angle increases again. Therefore, we do not believe that the pyramidalization of the carbon shared by D5 and D6 can be the driving force behind the photochemical process, but at most can be a consequence of the twisting.

3. Fig. 2, legend. It is not clear how the the Pr distribution has been generated in panel c.

Authors' Reply: We apologize if the strategy adopted in this work was not clear.

As commented before we have significantly revised the whole section of the Methods; in particular, we have added a specific paragraph to describe the sampling of the Pr. We have also revised the beginning of the subsection "The photoisomerization" and added the following comment in the caption of Fig.2:

"The Pr distribution has been generated from a 4 μ s-long MD simulation previously performed in our group."

4. Has the CI been optimized and characterized for some configurations and, thus, studied more carefully (e.g. is that a peaked or sloped CI)?

Authors' Reply: As the system (chromophore and environment) is quite complex, we refrained from doing optimizations at the stage of the photochemistry. Although we are not in measure to formally characterize the conical intersection as sloped or peaked, such information would only provide some insight into how accessible the conical intersection is. On the other hand, by our SH simulations, we are in measure to fully disclose this matter: in particular, the CI region is very easily accessed from the Franck Condon point, as all the trajectories approach it in a relatively short time after excitation.

Main text, page 4.

5. The unidirectional isomerization motion is caused by the chirality of the protein environment, and it is reminiscent of the well know purely unidirectional motion documented in certain rhodopsin proteins (especially bovine rhodopsin). However, I ask the authors to be careful and give their definition of counterclockwise and clockwise. In fact, such definition depends from the reference or convention adopted. As a matter of fact when adopting the convention adopted for bovine rhodopsin the present motion would be clockwise. This may confusing a reader not familiar with the specific convention used for bilin.

Authors' Reply: We thank the Reviewer for the suggestion. Our definition of counterclockwise is adapted from the phytochrome literature (10.1073/pnas.0902370106). In the revised text we have

revised Fig. 1 highlighting with a red arrow the direction of rotation, and in the Introduction, we have added the following sentence

“Such a CI is reached via a hula-twist mechanism, which involves a rotation of the D-ring in the counterclockwise direction according to the convention used in the literature [...] (see also the inset in Fig. 1).”

6. "...the type of construct..." -> "...type of computational model...."

Authors' Reply: We apologize for being unclear here. With this sentence, we were referring to the experimental results. For the sake of clarity, we revised the above sentence in the following way:

“The excited-state lifetimes are affected by the composition of the investigated phytochrome[...].”

7. "...To assess the robustness of the obtained decay times, we firstly repeated set A and set B SH trajectories, forbidding hops when the energy gap between electronic states S0 and S1 is larger than 0.5 eV....". It is not easy to understand the significance of this. Wouldn't be enough to plot in an histogram the distribution of the energy gaps at the hop points?

Authors' Reply: Indeed, we originally analyzed the distribution of energy gap, as the Reviewer suggests here, and we noticed that a fraction of hops indeed occurred above 0.5 eV. This is the reason why we decided to investigate their role and we rerun the simulations with the requirement on the energy gap to quantify the effects of forbidding the hops. This analysis would have been possible only by re-analyzing the SH trajectories.

8. "...of set A/B by artificially increasing the barrier around the dihedral D6, namely from the "real" one of 1.6 kcal mol⁻¹ to 5.2 kcal mol⁻¹ (Fig. S10)....". How is this barrier computed? Did the authors perform TS search or just a relaxed scan?

Authors' Reply: The barrier of 1.6 kcal/mol was computed by means of a relaxed scan (see Supplementary Figure 3). We first optimized the chromophore in the ground state and then we optimized each structure keeping the D6 dihedral frozen. Afterwards, we optimized each ground state-optimized structure at the excited state keeping the dihedral D6 frozen.

9. Fig. S12. The authors have to state which kinetic theory has been used to compute the mentioned lifetimes: TST? That would require to assume the thermal equilibration of a planar excited state.

Authors' Reply: The data points in Figure S12 (Supplementary Figure 13 of the revised manuscript) (circles) were obtained from SH simulations with added barriers. The population fittings shown in Figure S11 (Supplementary Figure 12 of the revised manuscript) gave us the excited-state lifetimes, which were then used in Figure S12 (Now supplementary figure 13). We did not use the TST since our trajectories are far from equilibrium. To analyze these data, we simply assumed that log(τ) was a linear function of the barrier height. Notably, the exact same slope fitted the data of both set A and set B. The star in Supplementary Figure 13 was obtained by extrapolating the data in Set A. We have added the following comment to the caption of Supplementary Figure 13:

“The population fittings shown in Supplementary Fig. 12 give us the excited-state lifetimes.”

10. "...the final ground state configurations of the SH reactive trajectories...". How is this configuration exactly defined?

Authors' Reply: We apologize if the set-up of the ground-state QM/MM MDs was not clear enough. We have changed the quoted sentence:

"The latest conformations sampled by the SH reactive trajectories were used as starting point to characterize the time evolution of the photoproduct through ground-state QM/MM and full MM MD simulations (see Methods)."

Main text pag. 5.

11. Fig. 3a. This is a minor point. It is not clear why the authors depict the conical intersection region with an avoided crossing (this is confusing). That would be OK if a real minimum would exist in the vicinity of a conical intersection (i.e. if the conical intersection would be sloped rather than peaked).

Authors' Reply: We thank the reviewer for pointing out that the figure can be misleading. We corrected it in the revised Figure 3a.

12. "... the propionyl group (D1B)h...". This has to be indicated in some of the chromophore schemes above.

Authors' Reply: We changed the sentence to explain the propionyl group dihedral:

"The only exception is represented by the first dihedral of the propionyl group attached to ring B (D1B, Supplementary Fig. 8), which is stabilized at Pfr-like values (Supplementary Fig. 14)."

Main text pag. 8

13. "Discussion" -> "Discussion and Conclusions"

Authors' Reply: We thank the Reviewer for noticing this typo: we corrected it in the revised text.

14. "... The different behavior is not determined by the type of chromophore but instead by the specific composition and shape...". This conclusion appears to be too strong. Is there a solid evidence for the irrelevance of the chromophore type and structure? When dealing with biological photoreceptors this type of statements are generally too strong to be true. Biological evolution has usually evolved both the chromophore and the cavity composition to optimize a certain function. These "parts" are mechanisms of a single machinery. I would modify these strong statements to remain on the safe side.

Authors' Reply: With the above-cited statement, we were specifically referring to the different types of bacterial phytochromes that contain biliverdin IX α as a chromophore. Phytochromes can bind biliverdin IX α , phytochromobilin or phycocyanobilin. Bacterial phytochromes share the same biliverdin IX α chromophore, but within such a family it is possible to have either prototypical or bathy phytochromes. According to the literature (10.1111/php.12742 – 10.1038/nchem.2225),

what makes the Pfr state thermodynamically more stable than the Pr state in bathy phytochromes are the different interactions with the environment.

Therefore, what determines the behavior of these phytochromes is definitely not the chromophore type (which is the same in all bacterial phytochromes) but the protein. Obviously, the protein is always adapted in order to bind biliverdin IX α and shape its properties. However, our reasoning still stands, in that the different protein pockets determine different behavior of the chromophore.

Main text pag. 9

15. "... As a matter of fact, our simulations confirm that the binding pocket has a fundamental role even beyond the initial photoisomerization process...". Again, the authors should not stress this factor which is a trivial one (just think to the color tuning in different photoreceptors, as well as the differences in behavior with respect to the solvent environment). Of course, what is really important here is the description of the actual molecular level interactions bringing about a certain behavior especially after the isomerization. This is, in a sense, a novelty in the field as it is not often explored especially on the microsecond timescale.

Authors' Reply: We agree that the role of the protein is already well recognized for color tuning and other properties of chromophores. However, here we are referring to the effect of the protein pocket after the photoisomerization has taken place, i.e., in the ground-state evolution of the system towards the final photoactivated state through the Lumi-R intermediates. This is, in our opinion, not trivial at all, because often we expect an action of the (photoisomerized) chromophore on the protein. Here, however, we find a strong indication that the protein can guide the further evolution of the system towards the correct final state. Our interpretation agrees with mutagenesis experiments, which show an incomplete photoconversion process upon substitution of important residues.

16. "... see Fig. S12...". There is a mismatch between the figure number and the actual reported data. I did not find such data in the Supporting Information.

Authors' Reply: We thank the Reviewer for noticing this typo: we corrected it in the revised text.

17. "...According to this scheme, the C3b carbon atom of the chromophore is the connection atom which behaves as a hydrogen atom in the QM part and as a normal carbon atom in the MM part (Fig. S7)....". Is this just the standard H-link atom method?

Authors' Reply: It is a very similar but not exactly identical method. The link-atom scheme saturates the valence of the QM part by adding a monoelectronic atom (usually a hydrogen atom) that is not part of the real system. In the connection atom scheme, the boundary atom (C3b) is replaced by a hydrogenoid atom with one *s* atomic orbital (*Theor Chem Acc (2004) 111:270-279*).

Main text, page 10.

18. "... were initially run for 20 ps using a time step $\Delta t = 0.1$ fs and the Bussi-Parrinello stochastic thermostat at 300K. The last 10 ps were used for sampling the initial conditions (nuclear coordinates

and momenta) ...". 20+10 ps is really far too short for a complete equilibration run. I guess that this is due to the fact that the authors use a QM/MM method that is probably (even if a semiempirical QM method is used) much more expensive than the MM method. A different strategy adopted in the literature would have been to equilibrate the system with MM for several hundred ns using a parametrized chromophore and then apply the QM/MM equilibration starting from the MM ns-equilibrated system. I wonder why this strategy has not been adopted.

Authors' Reply:

As reported previously, unfortunately, the description of the computational protocol was not clearly described in the original version of the manuscript, and this has generated some misunderstandings. In the revision we have significantly revised the whole section of the Methods; in particular, we have added a specific paragraph to describe the sampling of the Pr:

"In a previous study, we investigated the conformational space of the resting (Pr) using a 4 μ s-long MM-MD simulation [...]. Five different configurations were extracted from such dynamics and used as starting points for ground-state QM/MM MD trajectories. The latter were run for 20 ps starting using a time step $\Delta t = 0.1$ fs and the Bussi-Parrinello stochastic thermostat at 300K. To keep the computational cost feasible, QM/MM trajectories were run on a reduced system (Supplementary Fig. 19) and to avoid any evaporation of water molecules, we added a constraining potential. The BV chromophore was treated with the multi-reference floating occupation molecular orbital-complete active space configuration interaction (FOMO-CASCI) method [...] in combination with the AM1 Hamiltonian. An active space of (6,6) was used in combination with Gaussian width of 0.05 a.u. for FOMOs (details on the selection of the QM method are provided in the Supplementary Methods 1). Conversely, the protein was described with the AMBER ff14SB force field and water molecules with TIP3P. The boundary between the covalently linked QM and MM parts was treated using a connection atom scheme previously used for semiempirical QM/MM approaches. According to this scheme, the C3 β carbon atom of the chromophore is the connection atom which behaves as a hydrogen atom in the QM part and as a normal carbon atom in the MM part (Supplementary Fig. 8). QM/MM trajectories were performed with a modified version of the semiempirical program MOPAC2002 [...], interfaced with the TINKER software package [...]."

We have also revised the beginning of the subsection "The photoisomerization" to describe the H-bond features obtained in the long MM-MD simulation of the Pr state.

19. The set up of the the A and B runs of Pr needs to be described in more details.

Authors' Reply: We thank the Reviewer for pointing out to us that our discussion was not clear. We have revised the whole Methods section; in particular, the paragraph for the "Surface Hopping simulations" now reads:

"The last 10 ps of the ground state QM/MM trajectories were used to extract the initial conditions (nuclear coordinates and momenta) for generating the swarm of non-adiabatic surface hopping (SH) trajectories. In particular, two trajectories, A and B, were used to generate a total of 2442 SH trajectories classified as set A and set B respectively. The other ground state trajectories were used to generate 881 additional SH trajectories for an analysis of the H-bonding effects (Supplementary Note 1). In all SH trajectories, the QM and the MM subsystems, as well as their respective levels of theory, were the same used in the QM/MM ground state simulations. The local diabaticization algorithm [...] was used for the integration of the time-dependent Schrödinger equation for the electrons, and quantum decoherence was taken into account by applying an energy-based correction [...]. More

details are provided in the Supplementary Methods 2. SH simulations were performed with a modified version of the semiempirical program MOPAC2002 [..], interfaced with the TINKER software package [..]."

20. "...was used for the integration of the time-dependent Schrödinger equation for the electrons, and quantum decoherence was taken into account applying...". One sentence defining the SH method used needs to be inserted here.

Authors' Reply: See the answer to the previous point

21. "...As now the only investigated electrostatic state is the ground state, an HF/AM1 approach has been used to describe the chromophore in the QM/MM MD simulations....". The authors choose to use a semiempirical QM method rather than parametrized the chromophore. A justification/motivation for this shall be introduced here.

Authors' Reply: In the ground-state simulations of the photoproducts, we used AM1 to introduce as little discontinuity as possible with respect to the nonadiabatic (surface hopping) simulations. Finally, to extend the investigation of the time evolution in the μs scale, we used a MM description this timescale is not feasible with a QM/MM description even if using a semiempirical Hamiltonian. In any case, we checked, through a PCA analysis, that the QM/MM MDs had explored a subset of the MM MD configurations (see Supplementary Figure 6 in the revised manuscript).

22. "...There, the GAFF force-field was used for BV and for the bonding interactions between BV and the covalently bonded cysteine residue Cys24...". Is the GAFF force-field good enough? The chromophore could, in principle, be specifically parametrized. I wonder if this has been done by other groups working on phytochrome chromophores.

Authors' Reply: In the literature, there exist a few parametrizations of BV. One is based on the CHARMM force field (*J. Phys. Chem. B*, 2009, 113, 4, 945-958), whose functional form is different from AMBER, so it is not compatible with our protein force field. An AMBER-compatible force field was obtained in *J. Phys. Chem. B*, 2019, 123, 2325-2334, assigning atom types from the AMBER ff03 protein force field (instead of GAFF). In our protocol we used the same strategy but, since the BV molecule presents functional groups that are different from the protein, we opted for assigning the GAFF force field, which gives a good performance on organic molecules. Our GAFF parametrization of BV was based on a careful choice of atom types in order to conform to the conjugation pattern of the chromophore. This force field for BV was already used in previous work on both the (resting) Pr and (active) Pfr states of the phytochrome; the resulting ensembles successfully reproduced different spectroscopic properties of the system, indicating that GAFF (with our assignment of parameters) can indeed properly describe BV.

23. "...On such subsystem, we performed a ONIOM(QM:MM)82 geometry optimization, in which the QM part was treated at B3LYP+D3/6-311G(d,p) level of theory....". This is a kind of awkward procedure. Initially the QM/MM model used by the author was based on the link-atom scheme and

therefore an additive scheme. Now, suddenly, the author switch to a subtractive scheme. This is not justified. There is I think not good reason for this switching of “technology”. So, an explanation has to be provided. I mean I understand that for computing IR spectrum one has to improve the QM level of theory. However this can be done also within the consistent link-H atom additive scheme.

Authors’ Reply: As a matter of fact, the ONIOM formulation used here is based on the *same technology* used in the additive QM/MM approach of both the SH calculations and the following ground state trajectories of the photoproduct. In fact, also here we use an electrostatic embedding and a parallel treatment of the QM-MM cut using a link atom (see also the answer to point 17). The only difference is represented by the QM level of theory, AM1 in SH and in the QM/MM ground state dynamics and B3LYP in the geometry optimizations and IR spectra. This change is due to the well-known difficulties of semiempirical methods in accurately simulating IR spectra. This has been clarified in the revised version of the Methods Section:

“For each frame, we considered the protein and a shell of 390 water molecules centered on the chromophore and we performed a geometry optimization of the QM subsystem in a frozen environment represented as an electrostatic MM embedding. The QM part included the BV chromophore, the side chains of Cys24, His290 and Tyr263, and the closest water molecules within 3.2 Å of ring D carbonyl and amidic nitrogen. The Cys24 residue was cut at the CA-CB covalent bond (Supplementary Fig. 8). As QM level we used B3LYP+D3 in combination with 6-311G(d,p) basis set. For these calculations, we used the ONIOM(QM:MM) scheme, implemented in the Gaussian 16 suite of programs. More details about the calculation of the IR spectra are reported in the Supplementary Methods 5.”

Reviewer #2 (Remarks to the Author):

The paper "Protein control of photochemistry and transient intermediates in phytochromes" by Salvadori et al reports QM and MM simulations on a canonical bacteriophycophrome. The work covers the photoisomerization process and the first ground state intermediate (lumi-R). The authors report evidence for how the chromophore isomerizes, how the protein environment controls this isomerization, and how the protein and chromophore develop the signal in the lumi-R state. The report underpins a recent result (by IR spectroscopy) that there are two lumi-R states present. I find these results important, because they connect atomic structure, photochemistry and signaling in phytochromes, which are a widely debated topic and because it advances our knowledge in the area.

Although some of the ideas had already been mentioned as ideas and models by others, this is the first rigid theoretical investigation that sensibly explains the formation of the lumi-R state. The paper gives a very plausible explanation at the atomic level on how bacteriophycophromes get photoactivated. It has the potential to become a paradigm model for phytochromes for many years to come and is therefore of highest interest for the phytochrome community.

The work advances the understanding of how proteins control (photo)chemistry. This is relevant in many more fields: biochemistry, photosynthesis, and related fields.

Although I have some minor comments below, I am pleased with the current state of the paper. I believe that the simulations support the conclusions and claims. I am not an expert in QM/MM, and cannot judge the technical validity of the simulations, but judged against basic thermodynamic and photochemical knowledge, the conclusions are well supported by the data. The conclusions also align well with previous literature. The methodology and choice of simulations is sensible and appropriate. The work should be reproducible from the details provided. I recommend publication with minor revisions.

Authors' Reply: We sincerely thank the reviewer for the positive comments

Comments:

Fig 1: I find it hard to understand which parts of the figure come from the paper and are new and which not. Could you clarify/revise the figure? It would be good to indicate what is "this work" and what is "from literature".

Authors' Reply: We thank the Reviewer for pointing out to us that our figure was not clear. We revised the caption in the following way:

"...Structures of the chromophore and the nearby residues obtained in this work for the early and late Lumi-R intermediates are reported on the two upper corners....."

Time scales: Fig 3 is used to claim time scales of the transition. Could you discuss the validity and certainty of these time scales? Surely this must depend on the choice of force field/basis set/resting state structure? Spectroscopy indicates a long excited state lifetime of up to nanoseconds for the DrBphP. This is now observed in the simulations. Could you discuss this?

Authors' Reply: To the best of our knowledge, spectroscopy indicates excited-state lifetimes up to hundreds of picoseconds for DrBph. As we have detailed in the final part of the Photoisomerization part of the Results Section, the fact that our simulations give smaller lifetimes (also for the slowest set of trajectories) is mainly due to the underestimation of the energetic barrier to the torsion in the excited state when using the selected QM level. In fact, by repeating the simulations on representative subsets of set A/B by artificially increasing the barrier around the dihedral D6, we have clearly shown that the lifetime increases exponentially with the height of the barrier. This, qualitatively speaking, tells us that our representation of the system is physically correct and, with a more accurate description of the D6 barrier, we would indeed be able to obtain excited-state lifetimes in very good agreement with experiments (see Supplementary Figure 13 in the revised manuscript).

Related: How accurate is the determination of the time scale of transition from early to late Lumi-R? It appears to agree with the experimental value, but is the accuracy of the time constant not quite inaccurate from the MD simulations?

Authors' Reply: We understand the Reviewer's concerns about classical MD simulations. Of course, the timescales of MD simulations depend on the quality of the force field. However, we note that biomolecular force fields such as ff14SB are usually able to accurately predict timescales for conformational changes or even for protein folding (See *e.g. J. Chem. Phys.*, 2020, **153**, 185102 DOI: 10.1063/5.0022135).

The precise value of the transition time from early to late Lumi-R clearly depends on many variables. However, the time scale of about one microsecond is substantially different from the 88 microseconds observed for the Meta-R formation. This is sufficient for us to assign the observed evolution to the early-to-late Lumi-R transition.

Fig 4 and 6 - please indicate structure of the late and early intermediates. This would help to understand the figures.

Authors' Reply: The structures of early and late intermediate are shown in Figure 1 of the manuscript. In the revised manuscript we have changed the caption of the figure to clarify this point and we have explicitly referred to it in the main text.

The absence of any protonation/deprotonation reactions in the model deserve discussion. Heyne has recently demonstrated fast deprotonation in the Pfr excited state; Hildebrand, Mrogrinski esobar; Ihalainen; and others have a number of recent papers indicating the importance of a proton equilibria in the binding pocket, so protonation is seems to be important. That it is not observed in the simulations of the authors could be a limitation of the model. Could you discuss this issue?

Authors' Reply: The reviewer is right about the crucial role of the deprotonation/protonation mechanism in phytochromes' activation.

We suppose the reviewer refers to the recent articles in *Nature Chemistry* (10.1038/s41557-022-00944-x), *Phys. Chem. Chem. Phys* (10.1039/d2cp00020b), and *Photochem. Photobiol. Sci.* (10.1007/s43630-021-00090-2).

Although proton transfer processes are a general mechanistic property of all phytochromes, they are still far from being fully understood. However, the literature seems to agree that a proton transfer process has to take place prior to – or concomitant with – the secondary structure transition of the tongue during the transition from Meta-R to Pfr (canonical phytochromes) or from Meta-F to Pr (bathy phytochromes). These processes are not covered by the current work.

For bathy phytochromes, it seems that the deprotonation of the chromophore plays a crucial role also in the first photochemical process toward the Lumi-F intermediate (10.1038/s41557-022-00944-x), but the literature reports a different behaviour for the *Deinococcus radiodurans* phytochrome. Here, no proton transfer process is reported before the formation of the Lumi-R, which is characterized by a dynamic chemical environment that has been confirmed by our simulations. Therefore, we refrained from considering protonation changes, and we assumed that the chromophore was always protonated.

Could you add some discussions regarding the (potential) limitations of the force field with respect to the lumi-R structures? To me it seems that especially the torsion angles of the BV are strongly dependent on the choice of dihedral constraints. Groenhofs JPCL 2022 recent paper does not arrive at the same structure for Lumi-R and it is said that the present simulations are unbiased, whereas Groenhofs are not - but is this really true given that there always is a force-field involved in MD simulations? I think it is important to discuss the potential limitations for the method.

Authors' Reply: It is true that our MD simulation, as the one of Groenhof and co-workers, involves a force field. Our claim on unbiased simulation however refers to the fact that here we used “plain” MD to sample the microsecond dynamics of Lumi-R without imposing a reaction coordinate, i.e. without an *additional* bias on the possible path taken by the system. Instead, Groenhof and co-workers used umbrella sampling simulations, selecting *a priori* a reaction coordinate to describe the evolution. This is indeed a “biased” simulation in the enhanced sampling jargon. We refrained from focusing on a specific coordinate, given the complexity of the system and the relatively short (sub-microsecond) time scale in which we expected to observe the Lumi-R evolution. In this sense, our simulations are unbiased.

Regarding the choice of force field, this Reviewer partially shares the concerns of Reviewer #1. In the literature, there exist a few parametrizations of BV. One is based on the CHARMM force field (*J. Phys. Chem. B*, 2009, 113, 4, 945-958), whose functional form is different from AMBER, so it is not compatible with our protein force field. An AMBER-compatible force field was obtained in (*J. Phys. Chem. B*, 2019, 123, 2325-2334), assigning atom types from the AMBER ff03 protein force field. In our protocol we used the same strategy but, since the BV molecule presents functional groups that are different from the protein, we opted for assigning the GAFF force field, which gives a good performance on organic molecules. Our GAFF parametrization of BV was based on a careful choice of atom types in order to conform to the conjugation pattern of the chromophore. This force field for BV was already used in previous work on both the (resting) Pr and (active) Pfr states of the phytochrome; the resulting ensembles successfully reproduced different spectroscopic properties of the system, indicating that GAFF (with our assignment of parameters) can indeed properly describe BV.

Reviewer #3 (Remarks to the Author):

Phytochrome are a class of photoreceptor in plants, bacteria and fungi used to detect light. The initial photoisomerization process will result in the structural changes from the chromophore to the whole protein. Giacomo et al. tried to uncover the molecular mechanism for how the interaction between the chromophore and the protein residue controls the sequence structural changes. They use an integrated multiscale molecular dynamic simulations method and found that the photoisomerization of the chromophore proceeds through a hula twist mechanism. The results in this paper may give rise to the interest of related areas, and my detailed comments are as follows.

Authors' Reply: We sincerely thank the reviewer for the positive comments

1. In Fig. 2, they ran two sets of ground-state QM/MM MD trajectories denoted as A and B, respectively. Then, they just believe that the trajectory A better captures the distribution of the dihedrals of D5 and D6. However, this inclusion is insufficient to acquire without further simulation sets. They should run at least two or more sets to double-check their results.

Authors' Reply: We thank the reviewer for the question which has also been done by Reviewer #1 (point 1). Here we repeat the same answer

Indeed, we fully agree that this point (also touched by Reviewer #3) is an important one and, in fact, to further support our conclusions we have extended the analysis as detailed here below. However, before describing the new results we would like to clarify the computational protocol used in this work that, we admit, was not very clearly described in the original version of the manuscript. In a previous study (ref. Phys. Chem. Chem. Phys., 2020, 22, 8585-8594), we investigated the conformational space of the resting (Pr) through a 4 μ s-long MM MD. Before starting the surface hopping simulations, we refined the description of the chromophore's geometry by running two 20 ps-long ground-state QM/MM MD trajectories (A and B in the manuscript) starting from two different configurations extracted from the μ s-long MM MD. The last 10 ps of the QM/MM trajectories A and B were then used for sampling the initial conditions (nuclear coordinates and momenta) for the swarm of SH trajectories. A total of 2442 SH trajectories were generated and classified into two sets, set A and set B.

Following the Reviewer's suggestion, we have now performed 3 additional 20ps-long ground-state QM/MM MD simulations starting from different configurations of the μ s-long MM MD. Then, we generated 881 additional SH trajectories using the last 10 ps of the three new QM/MM MD simulations. We then combined all the trajectories (set A, set B, and the new 881 trajectories) and classified them based on the interactions between the D-ring and the closest residues at the starting geometry. By using a hierarchical agglomerative algorithm, we obtained 4 clusters. The most populated cluster 0 is characterized by a hydrogen bonding distance between D-ring carbonyl (O_D) and His290 and shows long lifetimes (this cluster contains the original set A). Instead, Clusters 1 and 2, where such interaction is weak or absent, have much shorter lifetimes (like Set B). Cluster 3, even though His290 is not at hydrogen bonding distance to O_D , shows excited state lifetimes of the order of Cluster 0. This behavior can be explained by noting that in this cluster, O_D is H-bonded to a water molecule, which also interacts with His290. We think that these further simulations highlight even more strongly the role of hydrogen-bonding interactions at O_D in determining the lifetime. These results are now reported in the Supplementary Material.

In the main text we have also revised the “Methods” section and the beginning of the subsection “Photoisomerization” in the Results section, and added the following comments (page 4):

“... In order to support this picture, we sampled additional ground-state QM/MM MDs from different points of the Pr ensemble, and generated a total of 881 SH trajectories. These trajectories were pooled with sets A and B and categorized on the basis of the initial conditions (Supplementary Note 1). We find that the average excited-state lifetimes clearly correlated with the presence of a hydrogen bond to O_D (Supplementary Fig. 4F). This hydrogen bond occurs either directly with His290 (e.g. what already observed onset A) or through a bridging water molecule.”

2. The detailed simulation system is ambiguous. Especially, whether the system contains water molecules around the chromophore. When water molecules are taken into account, the transition energy $S_0 \rightarrow S_1$ and the PES of the excited state S_1 may be altered. In addition, the photoisomerization starts configuration with a hydroxyl group which does not create the hydrogen bond. Is this hydroxyl group hydrogen-bonded to the water molecule?

Authors’ Reply: The chromophore is surrounded by several water molecules: the most important ones are pyrrole water, which coordinates the pyrrole rings A, B, and C, and another water molecule that coordinates the pyrrole ring D. The NTO analysis reported in Supplementary Figure 2 shows that the first excited state is characterized by a $\pi \rightarrow \pi^*$ “short-range” excitation, localized on the chromophore (at least this is true in the Franck Condon region). Therefore, the presence of a water molecule should not significantly affect the PES of S_1 , as it would in the case of a $n \rightarrow \pi^*$ excitation.

3. The excited-state lifetime was just estimated by the time evolution of the electronic state population. This method may be random in statistics, which strongly rely on the initial setting. Moreover, quantitatively measuring this value usually requires averaging over many sets of electronic excited state simulation. Can they thetically get a lifetime through the stimulated emission equation?

Authors’ Reply: We have investigated a non-radiative process; thus, we did not apply the stimulated emission equation. However, it is important to stress that the electronic state populations (reported in Fig 3) were obtained by averaging over thousands of trajectories (see Tab. S3 for the number of trajectories).

4. The whole calculation processes are rather complicated. For example, they do ground-state QM/MM with two sets first. Then, they extracted selected trajectories from these trajectories to use as the beginning setup for the exciting computation. In addition, specific trajectories must be chosen for the IR computations. Can the authors include a schematic diagram to clarify? The authors may also show that all the vibrational frequencies are positive for the IR simulations in the SI.

Authors’ Reply: As already commented in previous answers to Reviewer #1, unfortunately, the description of the computational protocol was not clearly described in the original version of the manuscript, and this has generated some misunderstandings. In the revision we have largely changed the Methods section by introducing a separate description for all the different steps: 1) sampling of the Pr state; 2) Surface Hopping simulations, 3) MD simulations of the intermediates and 4) Calculation of the IR spectra. We think that this significant revision is enough to give a clear and detailed

view of the whole computational strategy and we have preferred not to add a diagram. The frequencies, and all the source data underlying all the Figures, the Supplementary Figures, and Supplementary Tables are provided as a source data file as requested by the policy of the journal.

5. They calculated three single states but did not display the corresponding absorption spectrum. I am unable to assess the oscillation strength of $S_0 \rightarrow S_1$. As a result, they must evaluate a significant number of single states, such as twenty. This will provide additional information to determine how the photo is relaxed to the S_1 . Furthermore, the authors need to perform another excited state dynamics simulation, in which the initial absorption state is from S_0 to S_2 .

Authors' Reply: The oscillator strength for the S_0 -to- S_2 excitation is indeed small. Moreover, we know from experiments that the Q-band (S_0 -to- S_1 excitation) is the biologically relevant one and this is well separated from the Soret band (S_0 -to- S_2 and higher excitations). In light of all these evidences, we do not think that further simulations in which the initial absorption state is from S_0 to S_2 can lead to relevant results for the investigated process.

6. In the surface hopping trajectories, they use the semiempirical method (AM1). Also, they have performed a benchmark against the method of TDCAM-B3LYP. Even so, the red line (TDDFT) in Figure S3 is incomplete, in contrast to the blue line (AM1). They should run a test of surface hopping trajectories in the framework of TDDFT, in order to confirm the feasibility of AM1 in this system.

Authors' Reply: The complete graph is the following

As it can be seen, the AM1 description qualitatively reproduces (TD)DFT also outside the Franck Condon region. Obviously, at the quantitative level, there are differences due to the inadequacy of TDDFT in describing a twist around a double bond.

In the revised manuscript we have replaced the original figure 3 in the Supplementary Information with this one.

7. In the abstract, the mapping method description into method content reflect that the using method is too abstract to understand. They should describe the method with a more explicit statement, such as surface hopping dynamics simulation of AM1.

Authors' Reply: As suggested by this reviewer and reviewer #1 we have revised the abstract

REVIEWERS' COMMENTS

Reviewer #1 (Remarks to the Author):

After a carefully reading the author's reply and of the comments of the other reviewers and, after looking at the manuscript, I recommend publication subject to a very minor revision (I do not need to look at the manuscript again).

Here are my requests (the target is to facilitate the process of reproducing the author's results).

1) insert somewhere in the manuscript or in the supporting info the explanations (even in a shorter form) provided at points 7, 14, 15, 21 and 22. I believe this is important.

2) make sure the reference mention in the reply to point 17 is reported when discussing the QM/MM scheme.

3) I am afraid that the authors did not directly answer point 23. In my question I was asking about the motivation for changing the QM/MM model from a link-atom additive scheme into the ONIOM subtractive scheme in different parts of the research. I realize that the same QM and MM "ingredients" have been employed. However, in my own experience and also in principle, additive and subtracting schemes will never give identical answers. I thus believe that using them in different parts of the research introduces an inconsistency. Of course, I agree that there is no reason to believe that this "methodological" inconsistency leads to an incorrect scientific answers.

Reviewer #2 (Remarks to the Author):

The authors have appropriately addressed my concerns. I still feel that figure 1 is somewhat unintuitive, but I leave it to the authors and editor to make it more accessible. I think my concern is that it is not immediately clear from figure 1 what the paper adds to understanding the photocycle. Otherwise: congratulations on a great paper!

Reviewer #3 (Remarks to the Author):

The authors have addressed all previously raised concerns adequately. I recommend its publication in the current form.

Reviewer #1

After a carefully reading the author's reply and of the comments of the other reviewers and, after looking at the manuscript, I recommend publication subject to a very minor revision (I do not need to look at the manuscript again).

Here are my requests (the target is to facilitate the process of reproducing the author's results).

1) insert somewhere in the manuscript or in the supporting info the explanations (even in a shorter form) provided at points 7, 14, 15, 21 and 22. I believe this is important.

Authors' reply: We have inserted in the main text a brief explanation to all the requested points.

2) make sure the reference mention in the reply to point 17 is reported when discussing the QM/MM scheme.

Authors' reply: The reference in the reply to point 17 was already cited in the revised manuscript: it is reference 77

3) I am afraid that the authors did not directly answer point 23. In my question I was asking about the motivation for changing the QM/MM model from a link-atom additive scheme into the ONIOM subtractive scheme in different parts of the research. I realize that the same QM and MM "ingredients" have been employed. However, in my own experience and also in principle, additive and subtracting schemes will never give identical answers. I thus believe that using them in different parts of the research introduces an inconsistency. Of course, I agree that there is no reason to believe that this "methodological" inconsistently leads to an incorrect scientific answers.

Authors' reply: Maybe the reviewer has misunderstood this point but the ONIOM formulation including QM-MM electrostatic embedding is formally and numerically identical to an additive QM/MM model.

Reviewer #2

The authors have appropriately addressed my concerns. I still feel that figure 1 is somewhat unintuitive, but I leave it to the authors and editor to make it more accessible. I think my concern is that it is not immediately clear from figure 1 what the paper adds to understanding the photocycle. Otherwise: congratulations on a great paper!

Authors' reply: we have changed Figure 1 to make it clearer. In particular, we have clearly reported which are the structures determined in this work and what is known from the literature.

Reviewer #3

The authors have addressed all previously raised concerns adequately. I recommend its publication in the current form.